# Impedance-Frequency Response of Closed Electrolytic Cells

**DOI:** 10.3390/mi14020368

**Published:** 2023-01-31

**Authors:** José Juan López-García, José Horno, Constantino Grosse

**Affiliations:** 1Departamento de Física, Universidad de Jaén, Campus Las Lagunillas, Ed. A-3, 23071 Jaén, Spain; 2Departamento de Física, Universidad Nacional de Tucumán, Av. Independencia 1800, San Miguel de Tucumán 4000, Argentina

**Keywords:** Poisson-Nernst Planck equations, closed electrolytic cells, electrical impedance spectroscopy

## Abstract

The electric AC response of electrolytic cells with DC bias is analyzed solving numerically the Poisson–Nernst–Planck equations and avoiding the commonly used infinite solution approximation. The results show the presence of an additional low-frequency dispersion process associated with the finite spacing of the electrodes. Moreover, we find that the condition of fixed ionic content inside the electrolytic cell has a strong bearing on both the steady-state and the frequency response. For example: the characteristic frequency of the high-frequency dispersion decreases when the DC potential increases and/or the electrode spacing decreases in the closed cell case, while it remains essentially insensitive on these changes for open cells. Finally, approximate analytic expressions for the dependences of the main parameters of both dispersion processes are also presented.

## 1. Introduction

Complex impedance spectroscopy is one of the most powerful analytical tools in the characterization of electrolytic cells [1,2]. Roughly, two types of approximations have been used to relate the complex impedance to the macroscopic system parameters: (i) numerical solutions of theoretical models based on the transport processes in the system [3,4,5], and (ii) equivalent electric circuits [6,7,8]. Among the theoretical models proposed, those based on the Nernst–Planck flux and the Poisson equations are the most widely used [9,10,11,12,13,14,15].

In the simple case where the electrodes are perfectly blocking (ideally polarized) and the distance between electrodes tends to infinity (infinite solution approximation), the standard formulation of the PNP model usually works well in the high-frequency domain [16,17,18,19,20,21], but cannot describe complex low-frequency behaviors often found at a solid electrode–electrolyte interface [22,23]. An improvement of the infinite solution approximation is the open system model, in which the distance between electrodes is finite but the ionic concentration far from the electrical double layers formed near the electrodes is constant and equal to the initial electrolyte concentration inside the electrolytic cell. Although this model predicts the existence of two dispersion processes, the predicted results generally do not coincide with those experimentally observed. Because of these discrepancies, many improvements of the theoretical model were proposed such as finite ionic size, adsorption phenomena, adsorption−desorption processes, or the accumulation of ions near the interfaces in the absence of adsorption (fractional diffusion) [24,25,26,27].

The open system approach, in which the number of ions in the cell is not required to have a constant value, is valid in many experimental setups and applications. However, development in recent years of micro and nano devices, as well as the manufacture of energy storage devices in which the interest is to maximize the amount of energy per unit volume, implies that the number of ions necessary to form the electrical double layers in the system is not negligible, as compared to the total number of existing ions. In this situation, the open system approximation is no longer valid and it is necessary to revise the theoretical model to include this effect and quantify its possible impact on the system behavior [28].

In previous works [29,30], we analyzed the behavior of closed systems (in which the number of ions inside the cell is finite and constant) in the steady state. The main conclusions of these works was that the concentration at the center of a closed cell becomes a function of the electric potential of the electrodes and of their spacing, and that the surface density of the electrical double layers is limited by the total amount of ions in the system [31]. In this work, we extend the previous analysis to the AC small-signal impedance in the presence of DC bias. We numerically calculate and provide a physical interpretation of the dependence of the high-frequency dispersion on the surface potential and the electrode spacing, see also Barbero et al. [32]. We similarly discuss the strong increase of the real part of the impedance and the behavior of the imaginary part of said parameter at low frequencies. Moreover, we compare all our results to the open cell solutions. Finally, several approximate analytical expressions are proposed that reproduce quite accurately all these behaviors.

## 2. Theory

We consider an electrolytic cell with parallel-plane blocking electrodes separated a distance 2L. The cell is closed and filled with a binary symmetric electrolyte solution with valence z, concentration C (mol/m^3^), and diffusion coefficient *D*.

### 2.1. Equation System

We use a Cartesian reference system with the electrodes located at x=±L. In the framework of the standard electrokinetic model (point ions, constant permittivity value that does not depend on the local ionic concentrations or the proximity to solid interfaces, validity of the Einstein equation relating mobility and diffusion coefficient) the system behavior is determined by the PNP model that is composed of the following equation set:
(i).Poisson equation:(1)$$\varepsilon_{e}\frac{\partial^{2}\Psi (x,t)}{\partial x^{2}}=-zeN_{A}\left[c_{+}(x,t)-c_{-}(x,t)\right]$$
where $\varepsilon_{e}$ is the absolute permittivity of the electrolyte solution assumed to have a constant value, Ψ is the electric potential, t the time, e the elementary charge, and $c_{\pm }$ the local concentrations of the two ionic types.(ii).Nernst–Planck equations:(2)$$J_{\pm }(x,t)=-D\left\{\frac{\partial c_{\pm }(x,t)}{\partial x}\pm c_{\pm }(x,t)\frac{\partial }{\partial x}\left[\frac{ze\Psi (x,t)}{kT}\right]\right\}$$
where $J_{\pm }$ are the ionic fluxes (mol/(m^2^s), k is the Boltzmann constant, and T the absolute temperature.(iii).Continuity equations:(3)$$\frac{\partial J_{\pm }(x,t)}{\partial x}=-\frac{\partial c_{\pm }(x,t)}{\partial t}$$

### 2.2. Boundary Conditions

We set the potential reference at the central plane:(4)Ψ(x=0,t)=0
so that the potential values at the electrodes are:(5)$$\Psi (x=\mp L,t)\equiv \pm \Psi_{S}(t)=\pm \left|\frac{\Delta \Psi (t)}{2}\right|$$
where ΔΨ(t) is the potential difference applied to the electrodes. 

On the other hand, the blocking electrode condition implies that the ionic fluxes must vanish at the electrode-solution interfaces so that:(6)$$J_{\pm }(x=\pm L,t)=0\;\Rightarrow \;{\frac{\partial c_{\pm }(x,t)}{\partial x}|}_{x=\pm L}\pm c_{\pm }(x=\pm L,t)\frac{\partial }{\partial x}{\left[\frac{ze\Psi (x,t)}{kT}\right]}_{x=\pm L}=0$$

Finally, the closed cell condition requires that the total ionic numbers in the cell must remain constant:(7)$$\begin{array}{cc} 2LC & =\int_{-L}^{L}c_{\pm }(x,t)dx=\int_{-L}^{0}c_{\pm }(x,t)dx+\int_{0}^{L}c_{\pm }(x,t)dx\\ & =\int_{-L}^{0}c_{\pm }(x,t)dx+\int_{-L}^{0}c_{\mp }(x,t)dx=\int_{-L}^{0}\left[c_{+}(x,t)+c_{-}(x,t)\right]dx \end{array}$$

This last equation is the only one that is different for the case of closed and open cells, for which the corresponding boundary condition is: c(0,t)=C. Note, however, that this condition is strictly true only in the stationary state, but not the time-dependent one, because the diffusive flows of ions in or out of the cell, connected to the infinite reservoir, are not instantaneous. Nevertheless, we used this condition as is universally performed, because the error made should be negligible in view of the fact that it only depends on the weak AC potential amplitude, and not the strong DC amplitude.

### 2.3. Dimensionless Variables

We shall use the following dimensionless variables:(8)$$\bar{\Psi }=\frac{ze\Psi }{kT}$$
(9)$${\bar{c}}_{\pm }=\frac{c_{\pm }}{C}$$
(10)$$\bar{x}=\frac{x}{\lambda_{D}}$$
(11)$${\bar{J}}_{\pm }=\frac{\lambda_{D}J_{\pm }}{DC}$$
(12)$$\bar{t}=\frac{Dt}{{\left(\lambda_{D}\right)}^{2}}$$
where
(13)$$\lambda_{D}=\sqrt{\frac{kT\varepsilon_{e}}{2z^{2}e^{2}N_{A}C}}$$
is the Debye length and $N_{A}$ the Avogadro number. 

This allows to write the equation system as follows:(14)$$\frac{\partial^{2}\bar{\Psi }(\bar{x},\bar{t})}{\partial {\bar{x}}^{2}}=\frac{{\bar{c}}_{-}(\bar{x},\bar{t})-{\bar{c}}_{+}(\bar{x},\bar{t})}{2}$$
(15)$${\bar{J}}_{\pm }(\bar{x},\bar{t})=-\frac{\partial {\bar{c}}_{\pm }(\bar{x},\bar{t})}{\partial \bar{x}}\mp {\bar{c}}_{\pm }(\bar{x},\bar{t})\frac{\partial \bar{\Psi }(\bar{x},\bar{t})}{\partial \bar{x}}$$
(16)$$\frac{\partial {\bar{J}}_{\pm }(\bar{x},\bar{t})}{\partial \bar{x}}=-\frac{\partial {\bar{c}}_{\pm }(\bar{x},\bar{t})}{\partial \bar{t}}$$
while the boundary conditions take the form:(17)$$\bar{\Psi }(\bar{x}=0,\bar{t})=0$$
(18)$$\bar{\Psi }(\bar{x}=\mp L/\lambda_{D},\bar{t})=\pm {\bar{\Psi }}_{S}(\bar{t})$$
(19)$${\frac{\partial {\bar{c}}_{\pm }(\bar{x},\bar{t})}{\partial \bar{x}}|}_{\bar{x}=\pm L/\lambda_{D}}\pm {\bar{c}}_{\pm }(\bar{x}=\pm L/\lambda_{D},\bar{t}){\frac{\partial \bar{\Psi }(\bar{x},\bar{t})}{\partial \bar{x}}|}_{\bar{x}=\pm L/\lambda_{D}}=0$$
(20)$$\frac{2L}{\lambda_{D}}=\int_{-L/\lambda_{D}}^{0}\left[{\bar{c}}_{+}(\bar{x},\bar{t})+{\bar{c}}_{-}(\bar{x},\bar{t})\right]d\bar{x}$$

## 3. Equation System Solution

In order to solve Equations (14)–(16) together with boundary conditions (17)–(20), the problem is split into two parts. In the first, the stationary state (upper index 0) is considered with a polarization voltage applied to the electrodes and no ionic flow anywhere in the electrolytic cell. Under these conditions, the ionic concentrations follow the Boltzmann distribution:(21)$${\bar{c}}_{\pm }^{0}(\bar{x})=\frac{c^{0}(0)}{C}\exp\left(\mp {\bar{\Psi }}^{0}(\bar{x})\right)$$
and the boundary condition (20) becomes:(22)$$\frac{L}{\lambda_{D}}=\frac{c^{0}(0)}{C}\int_{-L/\lambda_{D}}^{0}\cosh\left({\bar{\Psi }}^{0}(\bar{x})\right)d\bar{x}$$

Moreover, in a recent work [29] it was shown that, for weak double layer coupling, the ionic concentration at the central plane has the following value:(23)$$\frac{c^{0}(0)}{C}=a-\sqrt{a^{2}-1}$$
where
(24)$$a=1+8{\left(\frac{\lambda_{D}}{L}\right)}^{2}\sinh^{4}\left(\frac{{\bar{\Psi }}_{S}^{0}}{4}\right)$$

Additionally, under these conditions, the differential capacitance of the double layer is the following:(25)$$C_{Dif}=\varepsilon_{e}\frac{d}{d\Psi_{S}^{0}}\left({\frac{d\Psi^{0}(x)}{dx}|}_{x=-L}\right)=\frac{\varepsilon_{e}S}{\lambda_{D}}\gamma \cosh\left(\frac{{\bar{\Psi }}_{S}^{0}}{2}\right)$$
where
(26)$$\gamma =\sqrt{\frac{c^{0}(0)}{C}}\left[1-\frac{\sinh\left({\bar{\Psi }}_{S}^{0}/2\right)\tanh\left({\bar{\Psi }}_{S}^{0}/2\right)}{\sqrt{{\left(L/\lambda_{D}\right)}^{2}+4\sinh^{4}\left({\bar{\Psi }}_{S}^{0}/4\right)}}\right]$$

In the case of an open cell, the ionic concentration at the central plane does not change with an applied electric potential since, by definition, it is connected at this plane to an infinite reservoir of electrolyte solution with concentration *C*. However, the total number of ions inside the cell increases so that the average concentration becomes, see Appendix A:(27)$$C_{average}=C\left[1+4\frac{\lambda_{D}}{L}\sinh^{2}\left(\frac{{\bar{\Psi }}_{S}^{0}}{4}\right)\right]$$

As for the differential capacitance, it is reduced to the classic Gouy–Chapman expression:(28)$$C_{Dif}=\frac{\varepsilon_{e}S}{\lambda_{D}}\cosh\left(\frac{{\bar{\Psi }}_{S}^{0}}{2}\right)$$

Finally, both for closed and open cells, Equations (23)–(28) show that all cell-size-related effects become negligible when $\lambda_{D}\ll L$, as expected [28,29], and that the behavior of both cell types becomes identical for $\Psi_{S}^{0}=0$.

In the second part, a weak signal is superposed over the stationary state so that all system parameters can be written as their stationary value plus a perturbation term:(29)$$\bar{\Psi }={\bar{\Psi }}^{0}+\delta \bar{\Psi }$$
(30)$${\bar{c}}_{\pm }={\bar{c}}_{\pm }^{0}+\delta {\bar{c}}_{\pm }$$
(31)$${\bar{J}}_{\pm }={\bar{J}}_{\pm }^{0}+\delta {\bar{J}}_{\pm }=\delta {\bar{J}}_{\pm }$$
Note that this is a strictly linear expansion leading to an AC current proportional to and a cell impedance independent of the AC voltage, unlike higher-order expansions where these assumptions do not hold [33,34].

Inserting Equations (29)–(31) into (14)–(16), and assuming that the perturbation is sufficiently weak so that only linear terms should be considered, leads to:(32)$$\frac{\partial^{2}\delta \bar{\Psi }(\bar{x},\bar{t})}{\partial {\bar{x}}^{2}}=\frac{\delta {\bar{c}}_{-}(\bar{x},\bar{t})-\delta {\bar{c}}_{+}(\bar{x},\bar{t})}{2}$$
(33)$$\delta {\bar{J}}_{\pm }(\bar{x},\bar{t})=-\frac{\partial \delta {\bar{c}}_{\pm }(\bar{x},\bar{t})}{\partial \bar{x}}\mp \delta {\bar{c}}_{\pm }(\bar{x},\bar{t})\frac{d{\bar{\Psi }}^{0}(\bar{x})}{d\bar{x}}\mp {\bar{c}}_{\pm }^{0}(\bar{x})\frac{\partial \delta \bar{\Psi }(\bar{x},\bar{t})}{\partial \bar{x}}$$
(34)$$\frac{\partial \delta {\bar{J}}_{\pm }(\bar{x},\bar{t})}{\partial \bar{x}}=-\frac{\partial \delta {\bar{c}}_{\pm }(\bar{x},\bar{t})}{\partial \bar{t}}$$
where Equations (33) and (34) can be combined using Equation (21):(35)$$\begin{array}{l} \frac{\partial^{2}\delta {\bar{c}}_{\pm }(\bar{x},\bar{t})}{\partial {\bar{x}}^{2}}\pm \frac{d{\bar{\Psi }}^{0}(\bar{x})}{d\bar{x}}\frac{\partial \delta {\bar{c}}_{\pm }(\bar{x},\bar{t})}{\partial \bar{x}}\pm \frac{d^{2}{\bar{\Psi }}^{0}(\bar{x})}{d{\bar{x}}^{2}}\delta {\bar{c}}_{\pm }(\bar{x},\bar{t})-\\ -{\bar{c}}_{\pm }^{0}(\bar{x})\frac{d{\bar{\Psi }}^{0}(\bar{x})}{d\bar{x}}\frac{\partial \delta \bar{\Psi }(\bar{x},\bar{t})}{\partial \bar{x}}\pm {\bar{c}}_{\pm }^{0}(\bar{x})\frac{\partial^{2}\delta \bar{\Psi }(\bar{x},\bar{t})}{\partial {\bar{x}}^{2}}=\frac{\partial \delta {\bar{c}}_{\pm }(\bar{x},\bar{t})}{\partial \bar{t}} \end{array}$$

In the case of an harmonic perturbation, all the variables of interest can be written as:(36)$$\delta {\bar{\Psi }}^{*}(\bar{x},\bar{t})=\delta {\bar{\Psi }}^{*}(\bar{x})\exp(i\bar{\omega }\bar{t})$$
(37)$$\delta {\bar{c}}_{\pm }^{*}(\bar{x},\bar{t})=\delta {\bar{c}}_{\pm }^{*}(\bar{x})\exp(i\bar{\omega }\bar{t})$$
where *i* is the imaginary unit, the upper index * indicates a complex magnitude, and $\bar{\omega }$ is the dimensionless angular frequency defined as:(38)$$\bar{\omega }=\frac{{{(\lambda }_{D})}^{2}}{D}\omega$$

Combining Equations (36) and (37) with (32) and (35) transforms these expressions into:(39)$$\frac{d^{2}\delta {\bar{\Psi }}^{*}(\bar{x})}{d{\bar{x}}^{2}}=\frac{\delta {\bar{c}}_{-}^{*}(\bar{x})-\delta {\bar{c}}_{+}^{*}(\bar{x})}{2}$$
(40)$$\begin{array}{l} \frac{d^{2}\delta {\bar{c}}_{\pm }^{*}(\bar{x})}{d{\bar{x}}^{2}}=i\bar{\omega }\delta {\bar{c}}_{\pm }^{*}(\bar{x})+{\bar{c}}_{\pm }^{0}(\bar{x})\frac{d{\bar{\Psi }}^{0}(\bar{x})}{d\bar{x}}\frac{d\delta {\bar{\Psi }}^{*}(\bar{x})}{d\bar{x}}\mp \\ \mp \frac{d\delta {\bar{c}}_{\pm }^{*}(\bar{x})}{d\bar{x}}\frac{d{\bar{\Psi }}^{0}(\bar{x})}{d\bar{x}}\mp \delta {\bar{c}}_{\pm }^{*}(\bar{x})\frac{d^{2}{\bar{\Psi }}^{0}(\bar{x})}{d{\bar{x}}^{2}}\mp {\bar{c}}_{\pm }^{0}(\bar{x})\frac{d^{2}\delta {\bar{\Psi }}^{*}(\bar{x})}{d{\bar{x}}^{2}} \end{array}$$
while the boundary conditions become:

The potential value at the central plane must vanish since it has been chosen for the potential origin:(41)$$\delta {\bar{\Psi }}^{*}(\bar{x}=0)=0$$

The potential values at the electrodes must be half the potential drop across the cell in view of the symmetry of the considered problem:(42)$$\delta {\bar{\Psi }}^{*}(\bar{x}=\mp L/\lambda_{D})=\pm \delta {\bar{\Psi }}_{S}$$

The ionic fluxes at the blocking electrodes must vanish:(43)$${\frac{d\delta {\bar{c}}_{\pm }^{*}(\bar{x})}{d\bar{x}}|}_{\pm L/\lambda_{D}}\pm \delta {\bar{c}}_{\pm }^{*}(\pm L/\lambda_{D}){\frac{d{\bar{\Psi }}^{0}(\bar{x})}{d\bar{x}}|}_{\pm L/\lambda_{D}}\pm {\bar{c}}_{\pm }^{0}(\pm L/\lambda_{D}){\frac{d\delta {\bar{\Psi }}^{*}(\bar{x})}{d\bar{x}}|}_{\pm L/\lambda_{D}}=0$$

The total numbers of ions in the closed cell must remain constant:(44)$$\int_{-L/\lambda_{D}}^{0}\left[\delta {\bar{c}}_{+}^{*}(\bar{x})+\delta {\bar{c}}_{-}^{*}(\bar{x})\right]d\bar{x}=0$$

Finally, the expression for the cell impedance can be determined. This magnitude relates the total current flowing through the cell with the potential drop across the electrodes:(45)$$I^{*}Z_{T}^{*}=\Delta \Psi^{*}=2\delta \Psi_{S}^{*}$$

The total electric current is the sum of conduction and displacement current terms:(46)$$I^{*}=zeN_{A}S\left[\delta J_{+}^{*}(x,t)-\delta J_{-}^{*}(x,t)\right]-\varepsilon_{e}S\frac{\partial }{\partial t}\left(\frac{\partial \delta \Psi^{*}(x,t)}{\partial x}\right)$$
where S is an arbitrary area across which the total current value is evaluated [35]. Since this value does not change along the cell, it can be easily evaluated for *x* = −*L* in view of Equation (6):(47)$$I^{*}=-\varepsilon_{e}S\frac{\partial }{\partial t}\left({\frac{\partial \delta \Psi^{*}(x,t)}{\partial x}|}_{x=-L}\right)$$
or, using dimensionless variables:(48)$$I^{*}=-\frac{kT\varepsilon_{e}SD}{ze{\left(\lambda_{D}\right)}^{3}}\frac{\partial }{\partial \bar{t}}\left({\frac{\partial \delta {\bar{\Psi }}^{*}(\bar{x},\bar{t})}{\partial \bar{x}}|}_{\bar{x}=-L/\lambda_{D}}\right)$$

Therefore, the complex impedance of the cell can be written as follows:(49)$$\frac{Z_{T}^{*}}{2}=\frac{\delta \Psi_{S}^{*}}{I^{*}}=-\frac{\lambda_{D}^{3}}{\varepsilon_{e}SD}\frac{\delta {\bar{\Psi }}_{S}}{i\bar{\omega }{\frac{d\delta {\bar{\Psi }}^{*}(\bar{x})}{d\bar{x}}|}_{\bar{x}=-L/\lambda_{D}}}$$

It is usual to normalize this expression dividing it by the resistance of the electrolyte solution:(50)$$R=\rho \frac{2L}{S}=\frac{kT}{2z^{2}e^{2}N_{A}CD}\frac{2L}{S}=\frac{2\lambda_{D}^{2}L}{\varepsilon_{e}DS}$$
where ρ is the solution resistivity. This leads to: (51)$$Z=\frac{Z_{T}^{*}}{2R}=i\frac{\lambda_{D}}{2\bar{\omega }L}\frac{\delta {\bar{\Psi }}_{S}}{{\frac{d\delta {\bar{\Psi }}^{*}(\bar{x})}{d\bar{x}}|}_{x=-L/\lambda_{D}}}$$
which can also be written as:(52)Z=λDδΨ¯S2ω¯LIm(dδΨ¯*(x¯)dx¯|x¯=−L/λD)+iRe(dδΨ¯*(x¯)dx¯|x¯=−L/λD)Re2(dδΨ¯*(x¯)dx¯|x¯=−L/λD)+Im2(dδΨ¯*(x¯)dx¯|x¯=−L/λD)

## 4. Results

In this section, we present the most relevant results of the numerical solutions of the PNP equations for the considered closed system with blocking electrodes.

The numerical calculations were performed using an algorithm based on the network simulation method, which consists of modeling a physical process by means of a graphical representation analogous to circuit electrical diagrams, which is analyzed using an electric circuit simulation program such as PSpice. A full account of the network model used in this work is given in Ref. [5]. It must be noted that since the electric potential changes rapidly near the interface x=±L, an appropriate simulation space grid must be modeled. In this work, the *x*-space grid is automatically adapted to the evolution of the electric potential profiles. If, in the course of the simulation, strong changes of the dimensionless electric potential with *x* are detected in any x coordinate region, more grid points are added into that region. Appropriate simulation space grids were calculated in this way to ensure good accuracy. 

We first consider the stationary state, and then the system response to a weak AC signal.

### 4.1. Steady State

The finite ionic content in the electrolytic cell condition has a strong bearing on the steady-state behavior. As shown in previous works, it sets an upper limit on the double layer charge, which modifies the dependences of the surface charge and the differential capacitance on the electrode potential [29,30]. We shall here summarize those results that have the strongest impact on the frequency response. 

#### 4.1.1. Ionic Concentrations at the Central Plane of Closed Electrolytic Cells

When a potential difference is applied to a closed electrolytic cell, the ions move towards the electrodes to form the double layers. This modifies the ionic concentrations at the cell center, $c^{0}(0)$, so that this value is lower than the initial concentration C. The relative difference between these magnitudes increases when the initial concentration of ions is lowered, when the electrode spacing is lowered, or when the applied voltage is increased [28,29]. In view of approximate analytical solution (23), this quantity only depends on two parameters: the surface potential and the relation between the electrode spacing and the Debye length. These dependences are shown in Figure 1, where the $c^{0}(0)/C$ quotient is represented as a function of the surface potential for different values of $L/\lambda_{D}$. This figure also includes approximate analytical results obtained using Equation (23). Note that this being a theoretical work, the dimensionless surface potentials in this and the following Figures extend to very high values. While such values of the order of 400 mV are usually unrealistic, they were used in order to better understand the qualitative behavior of the plotted magnitudes.

As expected, $c^{0}(0)$ is always equal to C for ${\bar{\Psi }}_{S}^{0}=0$, and its value decreases with the surface potential. This decrement is significant when the number of ions required to form the double layers is comparable to the total number of ions of each sign inside the electrolytic cell:(53)$$N_{T}=2\;L\;S\;C\;N_{A}$$

For instance, for a dimensionless surface potential of 8 (approximately 200 mV), Figure 1 shows that this effect becomes important for a ratio $L/\lambda_{D}$ lower than 100. This figure also shows very good agreement between the approximate analytical solution, Equation (23), and the numerical results.

The behavior predicted by Equation (23) at high potentials
(54)$$\frac{c^{0}(0)}{C}\to {\left(\frac{L}{\lambda_{D}}\right)}^{2}\exp\left(-{\bar{\Psi }}_{S}^{0}\right)$$
is clearly observed in Figure 1.

#### 4.1.2. Electric Double Layer Thickness

The ionic concentration decrement at the central plane of a closed electrolytic cell leads to an increment in the electric double layer thickness. Figure 2 shows numerical results of the electric potential profiles normalized to the corresponding surface potential values. It is well known [33] that, according to the PNP model used in this work, the double layer thickness decreases monotonously with the surface potential value in the case of infinite spacing. This qualitative behavior can also be observed in Figure 2 for the lowest dimensionless potentials 2, 4, and 6, but clearly not for the highest potential values.

While it is not possible to strictly define the double layer thickness, it is common practice to identify it with the Debye screening length $\lambda_{D}$, Equation (13). For infinite electrode spacing and low surface potential values, this magnitude corresponds to the distance from the interface at which the potential decreases by a factor of e (base of the natural logarithms). Using this criterion in the general case allows the calculation of the double layer thickness using the condition: (55)$${\bar{\Psi }}^{0}\left(x=-L+L_{DL}\right)=\frac{{\bar{\Psi }}_{S}^{0}}{e}$$

For infinite electrode spacing, a rigorous expression for the electric potential profile exists [36], which can be combined with Equation (55) to give:(56)$$L_{DL}^{\infty }=\lambda_{D}\ln\left[\tanh\left(\frac{{\bar{\Psi }}_{S}^{0}}{4}\right)/\tanh\left(\frac{{\bar{\Psi }}_{S}^{0}}{4e}\right)\right]$$

Assuming that for finite spacing values there is no double layer overlap, the potential profiles approximately coincide with those corresponding to infinite spacing, making it possible to generalize Equation (56) to finite systems:(57)$$L_{DL}=\sqrt{\frac{kT\varepsilon_{e}}{2z^{2}e^{2}N_{A}c^{0}(0)}}\ln\left[\tanh\left(\frac{{\bar{\Psi }}_{S}^{0}}{4}\right)/\tanh\left(\frac{{\bar{\Psi }}_{S}^{0}}{4e}\right)\right]=\sqrt{\frac{C}{c^{0}(0)}}L_{DL}^{\infty }$$

This expression is identical to Equation (56), except for the bulk concentration *C* that has been replaced by the concentration at the central plane $c^{0}(0)$, which can be analytically evaluated using Equation (23). Note, finally, that in using the above-mentioned assumptions, the double layer thickness for open cells has the same value, $L_{DL}^{\infty }$, Equation (56), as for infinite spacing.

Figure 3 shows the electric double layer thickness values defined as functions of the surface potential and calculated for different values of $L/\lambda_{D}$. The solid lines correspond to numerical solutions of Equation (55), the dashed line to the infinite spacing solution, Equation (56), and the symbols to the approximate analytical solution, Equation (57). As can be seen, the solution corresponding to infinite spacing and open cells, dash line, is the only one that monotonously decreases with increasing surface potential and is independent of the parameter $L/\lambda_{D}$, Equation (56). For closed finite cells, this qualitative behavior only holds for small potential values, since for sufficiently high potentials a minimum is attained and the double layer thickness starts to increase. 

The double layer thickness minima occur where the first derivative of $L_{DL}$, with respect to the surface potential, vanishes. For $L/\lambda_{D}\gg 1$ and assuming high surface potentials, the minima of the electric double layer thickness curves are attained for, see Appendix A:
(58)$${\bar{\Psi }}_{S}^{0}\approx 2\ln\left(\frac{2}{e}\frac{L}{\lambda_{D}}\right)$$

The potential values so obtained, represented by vertical dot lines in Figure 3, show an excellent agreement with the minima of the numerically calculated curves.

In the extreme case, $L/\lambda_{D}=3$, the number of ions in the cell is so small that the thickness minimum cannot be seen in Figure 3. Finally, this figure shows that the approximate analytical expression (57) leads to a very good agreement with numerical results, except when double layer coupling starts to be significant.

Note that the effect of the bias voltage on the double layer overlap is opposite in open and closed cells. Consider, for example, the case of $L/\lambda_{D}=10$ and a potential value ${\bar{\Psi }}_{S}^{0}=8$ in Figure 3. For an open cell, dash line, the bias potential decreases the double layer thickness by a factor of approximately 0.4, so that the half-electrode spacing to double layer thickness ratio increases from 10 to 25. On the contrary, for a closed cell, light brown line, and dots, the double layer thickness increases by a factor of 2, so that the considered ratio decreases from 10 to 5 approximately. Because of this, the non-overlapping double layer hypothesis ceases to be strictly true, and the approximate solution starts to deviate from the numerical one in Figure 3.

### 4.2. Frequency Response

In this section, we discuss the frequency response of the system by means of an analysis of its complex impedance. Figure 4 shows the spectra of the real part of the impedance (4a), the imaginary part (4b), and the imaginary part multiplied by the angular frequency (4c), calculated for different $L/\lambda_{D}$ values, and a dimensionless surface potential ${\bar{\Psi }}_{S}^{0}=8$. Solid and dash lines correspond to closed and open cells, respectively, which all converge for $L/\lambda_{D}\to \infty$, as expected. In Figure 4c, the number of considered $L/\lambda_{D}$ values was reduced for the sake of clarity.

For both cell types, Figure 4a shows two dispersion processes associated to the two characteristic lengths present in the system. The low-frequency one is related to the spacing of the electrodes, and can be approximately determined as [17]:(59)$${\bar{\omega }}_{Low}={\left(\frac{\lambda_{D}}{L}\right)}^{2}$$
while the high-frequency term is related to the double layer thickness, and can be approximately computed as the following:(60)$${\bar{\omega }}_{High}=\frac{c^{0}(0)}{C}$$

Both dispersions can be clearly appreciated, except for the extreme cases $L/\lambda_{D}\to \infty$ and $L/\lambda_{D}=10$. In the first, this happens because when L→∞, Equation (59) shows that ${\bar{\omega }}_{Low}\to 0$, so that the dispersion shifts outside the plotted frequency range. In the second, when the electrode spacing and the double layer thickness become comparable, the double layers start to overlap, in which case the low and high-frequency dispersions are no longer distinct. This clearly occurs already in closed cells, but still not in open ones, due to their previously noted lower double layer thickness. 

The value of the real part of the impedance at frequencies between both dispersions can be estimated as being proportional to the resistance of the central part of the electrolytic cell, between the two double layers. Taking into account that the plotted values correspond to magnitudes normalized to the initial zero potential state, this resistance can be written in the case of closed cells as, see Appendix A:(61)Re(Z)=Cc0(0)(1−Cc0(0)λDL)

The values so obtained for the different considered cases, except for $L/\lambda_{D}=10$, correspond to the horizontal dot lines in Figure 4a. As can be seen, there is an excellent agreement between Equation (61) and the numerical results in the frequency range between both dispersions. As for $L/\lambda_{D}=10$, Equation (61) leads to a negative result, which is not surprising in view of the double layer coupling which invalidates the analytical solution, Equation (23), on which Equation (61) is based. 

In the case of open cells, the concentration at the central plane is always equal to the initial concentration *C*, so that Equation (61) transforms into: (62)Re(Z)=1−λDL

This expression, when compared to Equation (61) explains both the small amplitude of the high-frequency dispersion in open cells (the first factor $C/c^{0}(0)$ is absent in Equation (62)) and its weak dependence on the electrode spacing 2L (the second $C/c^{0}(0)$ factor is also absent and the $\lambda_{D}/L$ quotient is always small).

Figure 4b shows that the high-frequency dispersion in closed cells occurs at normalized frequency values close to 1 when the concentration at the central plane is approximately equal to the initial concentration (high $L/\lambda_{D}$ values). However, for increasing differences between $c^{0}(0)$ and C, that is, for decreasing $L/\lambda_{D}$ values, the characteristic frequency of this dispersion decreases in accordance with Equation (60). This can be clearly appreciated in Figure 4b, where the vertical dot lines correspond to solutions of that equation. Moreover, the excellent agreement between these lines and the maxima of the numerically computed curves can also be seen. As already mentioned, the sole exception corresponds to the $L/\lambda_{D}=10$ case that does not show a maximum because both dispersions are no longer independent of one another. 

As for open cells, the Debye length does not depend on the electrode spacing 2L, since the concentration at the central plane is always equal to C. Therefore, the normalized frequency corresponding to the high-frequency dispersion, Equation (60), is always equal to one. This behavior can also be clearly appreciated in Figure 4b.

Finally, Figure 4c shows spectra of the imaginary part of the impedance multiplied by the angular frequency. As can be seen, this product tends to a constant value at low frequencies, which is proportional to the inverse of the double layer differential capacitance, $C_{Dif}$, divided by the bulk capacitance, *C_B_*:(63)limω¯→0[ω¯Im(Z)]=CBCDif=1C¯Dif

This value can be evaluated analytically, taking into account that the bulk capacitance is:(64)$$C_{B}=\frac{\varepsilon_{e}S}{L-L_{DL}}$$
while the differential capacitance of the double layer is given in Equation (25) for closed and Equation (28) for open cells. The so-obtained values corresponding to closed cells are represented in Figure 4c by horizontal dot lines. As can be seen, there is a very good agreement between the analytical results and the numerical values as long as there is no double layer coupling, which is not the case for $L/\lambda_{D}=10$. As for open cells, Equation (28) shows that the corresponding differential double layer capacitance is always larger than for closed cells, Equation (25), which occurs because the double layer thickness is lower in view of the constant concentration value at the central plane. Moreover, since the bulk capacitance, Equation (64), does not depend on the ionic concentration, it follows that the limiting low-frequency ωIm(Z) values, represented in Figure 4c by horizontal dash-dot lines, should be always lower for open than for closed cells. This is clearly the case.

Figure 5 shows the spectra of the real part of the impedance (5a), its imaginary part (5b), and its imaginary part multiplied by the angular frequency (5c), calculated for different dimensionless surface potential values and $L/\lambda_{D}=100$. Solid and dash lines correspond to closed and open cells, respectively, which all converge for ${\bar{\Psi }}_{s}^{0}=0$, as expected. In Figure 5c the number of considered surface potential values was reduced for the sake of clarity.

As can be seen in Figure 5a, the characteristic frequency of the low-frequency dispersion is independent of the polarization potential value of the electrodes, both for closed (solid lines) and for open (dash lines) cells, in agreement with Equation (59). Moreover, this equation correctly predicts a dimensionless frequency of the order of $10^{-4}$ for the chosen $L/\lambda_{D}$ value. Finally, all the open cell spectra converge for low frequencies in view that the corresponding amplitude only depends on the $L/\lambda_{D}$ value, Equation (62), which is kept constant in Figure 5.

On the contrary, the characteristic frequency of the high-frequency dispersion diminishes with increasing potential values for closed cells. This behavior, observed but not physically justified in [32], is related to the corresponding decrease of the ionic concentration $c^{0}(0)$ at the central plane, Figure 1. Figure 5b shows the predicted frequency values obtained using Equation (60) by means of vertical dot lines. As can be seen, these lines show a remarkable agreement with the maxima of the imaginary part of the impedance that were numerically obtained. As for open cells, the normalized frequency corresponding to the high-frequency dispersion remains equal to one independently of the applied potential value in view of the $c^{0}(0)=C$ condition and Equation (60). 

Finally, Figure 5c shows spectra of the imaginary part of the impedance multiplied by the angular frequency and their dependence on the surface potential. Just as in Figure 4c, these spectra tend at low frequencies toward constant values that are proportional to the bulk capacitance, $C_{B}$, divided by the double layer differential capacitance $C_{Dif}$. For closed cells (solid lines) this dependence is non-monotonous: from ${\bar{\Psi }}_{s}^{0}=0$ to ${\bar{\Psi }}_{s}^{0}=9$, the considered quotient decreases, while it increases for higher surface potential values. This behavior is due to the non-monotonous dependence of the differential surface capacitance of closed cells on the surface potential, Equation (25). On the contrary, for open cells (dash lines), the differential surface capacitance increases monotonously with the surface potential, Equation (28). This produces the strong decrease of the low-frequency value of the considered quotient in open cells. Finally, the dot lines in Figure 5c correspond to the calculated $C_{B}/C_{Dif}$ values corresponding to closed cells. The agreement with numerical results is fairly good. 

## 5. Conclusions

In conclusion, we numerically solved the linear AC response of closed electrolytic cells with DC bias and also obtained approximate analytical expressions that allow to reproduce the most important characteristics of the numerical simulations. Our results differ qualitatively and quantitatively from the open cell solution, usually assumed to be correct when the electrode spacing is sufficient to prevent double layer overlap. 

The AC response of these two configurations only coincides when the electrode spacing is much larger than the Debye length or when there is no DC bias potential. In the general case, the differences appear because the ionic concentration at the central plane in closed cells diminishes with the electrode potential value in view of the condition of fixed total ionic content inside the cell, while this concentration remains constant for open cells in view of their contact with an infinite bath. In both cases, the spectra exhibit two dispersions: a high-frequency one dependent on the ionic concentration at the central plane and a low-frequency one dependent on the Debye length to electrode spacing ratio.

Thus, while the high-frequency dispersion process in open systems only depends on the electrolyte concentration inside the cell, in closed systems the characteristic frequency of this dispersion decreases significantly when the DC potential increases and/or the electrode spacing decreases. On the other hand, the real part of the normalized impedance at frequencies in between both dispersion processes in open systems is approximately equal to 1, regardless of the system parameters while, for closed systems, the real part of the impedance strongly depends on the electrode spacing to Debye length ratio and on the bias electrode potential. Finally, in open systems, the amplitude of the imaginary part of the low-frequency dispersion multiplied by the frequency decreases monotonically with the electrode potential while, for closed systems, it decreases down to a minimum and then starts to increase. 

## Figures and Tables

**Figure 1 micromachines-14-00368-f001:**
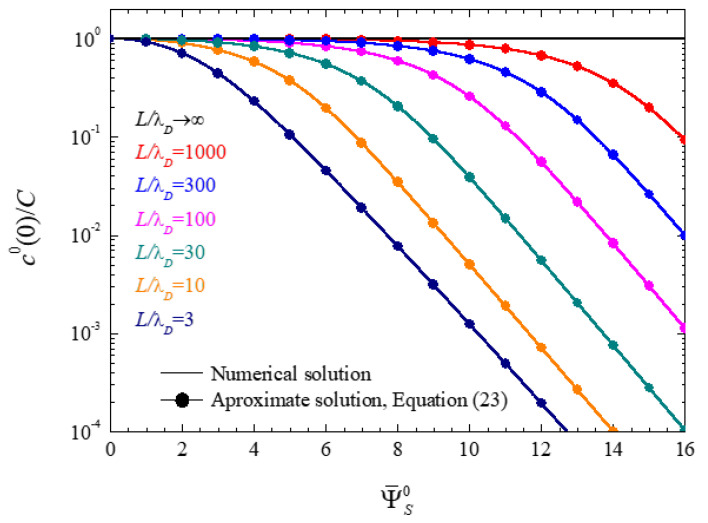
Ionic concentration at the central plane of closed electrolytic cells as function of the surface potential for the indicated $L/\lambda_{D}$ values.

**Figure 2 micromachines-14-00368-f002:**
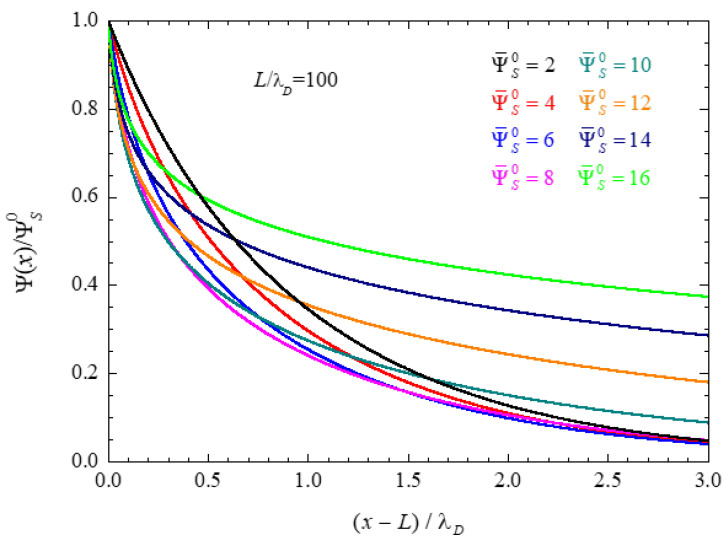
Numerical results for the electric potential profiles in a closed electrolytic cell calculated for $L/\lambda_{D}=100$ and the indicated surface potential values.

**Figure 3 micromachines-14-00368-f003:**
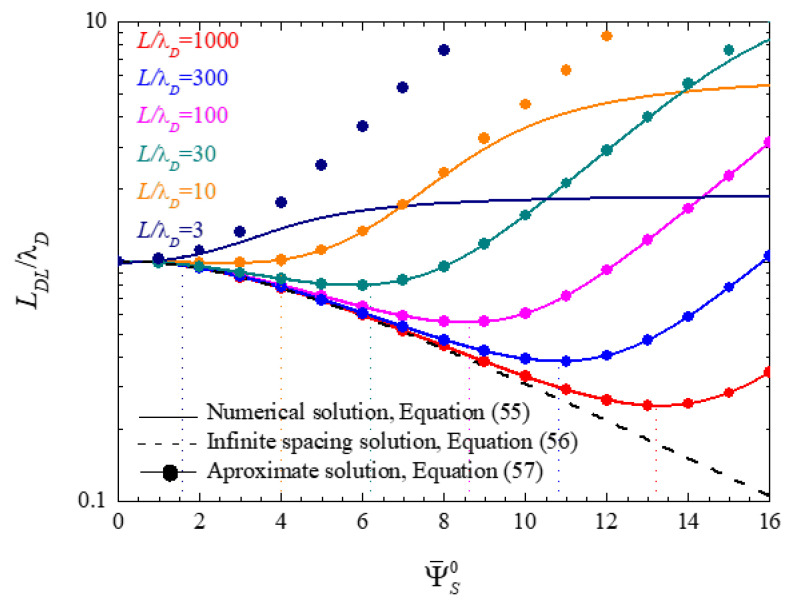
Dependence of the electric double layer thickness values on the surface potential, calculated for the indicated $L/\lambda_{D}$ values. Vertical dot lines show surface potential values corresponding to the double layer thickness minima, Equation (58).

**Figure 4 micromachines-14-00368-f004:**
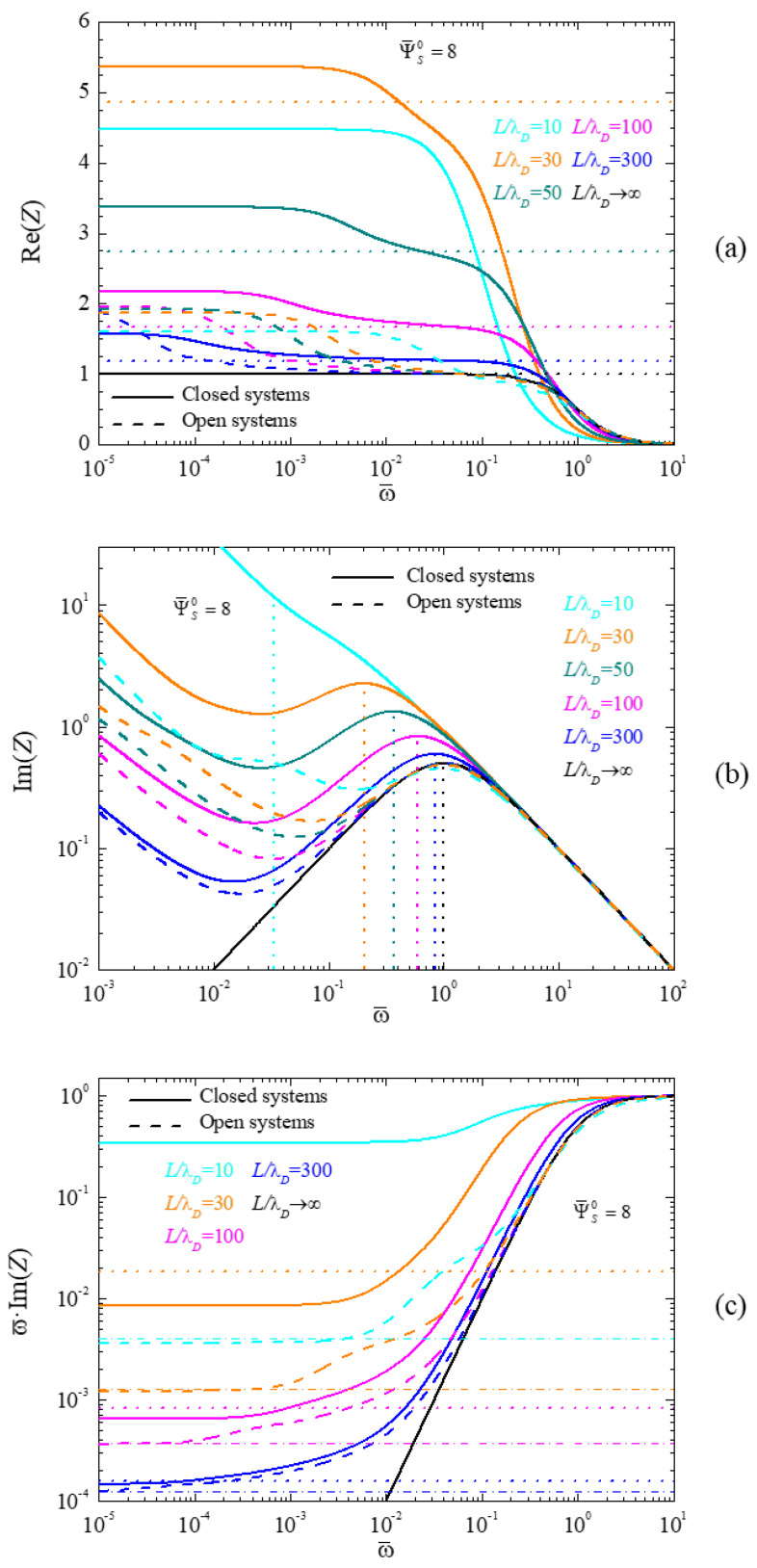
Spectra of the real part of the impedance (**a**), its imaginary part (**b**), and its imaginary part multiplied by the angular frequency (**c**), calculated for the indicated $L/\lambda_{D}$ values and the dimensionless surface potential ${\bar{\Psi }}_{S}^{0}=8$. Dot and dash-dot straight lines (closed and open cells, respectively) correspond to the following: (**a**) Real part of the impedance at frequencies between both dispersions, Equation (61) for open or (62) for closed cells; (**b**) characteristic frequency of the high-frequency dispersion, Equation (60) for closed cells; (**c**) Low-frequency limit of the imaginary part of the impedance multiplied by the frequency, Equations (63), (64), and (25) for closed or Equation (28) for open cells.

**Figure 5 micromachines-14-00368-f005:**
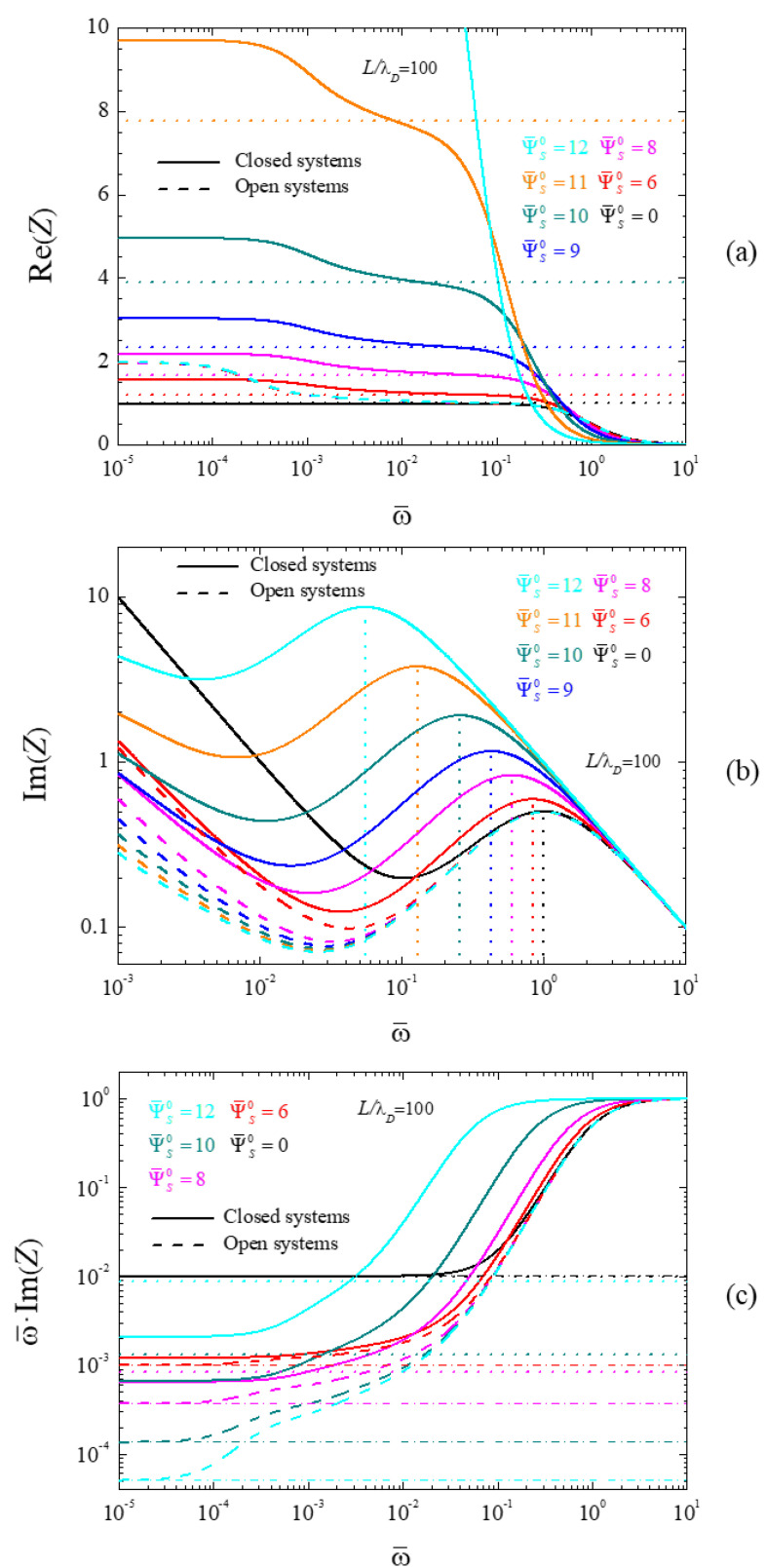
Spectra of the real part of the impedance (**a**), the imaginary part (**b**), and the imaginary part multiplied by the angular frequency (**c**), calculated for the indicated dimensionless surface potential values and for $L/\lambda_{D}=100$. Dot and dash-dot straight lines (closed and open cells respectively) correspond to the following: (**a**) Real part of the impedance at frequencies between both dispersions, Equation (61) for open or (62) for closed cells; (**b**) characteristic frequency of the high-frequency dispersion, Equation (60) for closed cells; (**c**) low-frequency limit of the imaginary part of the impedance multiplied by the frequency, Equations (63), (64), and (25) for closed or Equation (28) for open cells.

## Data Availability

The data used to support the findings of this study are available from the corresponding author upon request.

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
