# Peer review of "Impedance-Frequency Response of Closed Electrolytic Cells"

_micromachines, 2023, doi:10.3390/mi14020368_

Round 1
Reviewer 1 Report
The authors explore the properties of the Impedance-Frequency Response of Closed Electrolytic Cells. They model the cell filled with a symmetric electrolyte using the Poisson-Nernst-Planck (PNP) equations, which they solve numerically and analytically with the help of relevant approximations.
My first concern is the presentation of the relevant literature results. Positions [1-8] are not present in the reference section, and positions [29,30] do not have authors of the current contribution listed, which authors claim they should have.
My second concern is that the authors claim to have solved the problem numerically. Unfortunately, nowhere in the text, I found which equations were solved numerically and with what methods, software, numerical schemes etc. Therefore I cannot judge its correctness, and it is impossible to recreate the presented results to find if they are correct or not.
I also have doubts about problem formulation. Authors mix ionic concentrations and chemical potential. It is very hard to deduce what they really solve for. Moreover, normalization with the time-changing values (eq. 3 and eq. 5) is confusing. It would be very useful to write down full PNP equations for ionic concentrations and boundary conditions to understand better whether new terms appear there due to differentiations. The full PNP problem means all c+, c- and \Psi equations and boundary conditions without using \mu. In such a way, the picture will be more transparent.
Another major concern is that nowhere is the validity of the PNP approximation discussed. Figures suggest that potentials of 20fold thermal energy are applied, which requires at least some discussion. Here it is worth pointing also the notation. Authors seem to choose small y for dimensionless potential instead of Psi with an overbar, which they did for other variables. This is also confusing.
Authors need to thoroughly address those issues before publication.
Author Response
The referees' comments and suggestions (regular font) and our answers (italics) follow:
Reviewer-1
Comments and Suggestions for Authors
- The authors explore the properties of the Impedance-Frequency Response of Closed Electrolytic Cells. They model the cell filled with a symmetric electrolyte using the Poisson-Nernst-Planck (PNP) equations, which they solve numerically and analytically with the help of relevant approximations.
My first concern is the presentation of the relevant literature results. Positions [1-8] are not present in the reference section, and positions [29,30] do not have authors of the current contribution listed, which authors claim they should have.
This unfortunate problem is just a result of the automatic transformation of our submitted manuscript to the Micromachines format. This has been solved in the revised manuscript.
- My second concern is that the authors claim to have solved the problem numerically. Unfortunately, nowhere in the text, I found which equations were solved numerically and with what methods, software, numerical schemes etc. Therefore, I cannot judge its correctness, and it is impossible to recreate the presented results to find if they are correct or not.
The following paragraph was added to the manuscript in order to clarify this point:
The numerical calculations were performed using an algorithm based on the network simulation method, which consists in modeling a physical process by means of a graphical representation analogous to circuit electrical diagrams, which is analyzed using an electric circuit simulation program such as PSpice. A full account of the network model used in this work is given in Ref. [5]. It must be noted that since the electric potential changes rapidly near the interface x=+-L, an appropriate simulation space grid must be modeled. In this work, the x-space grid is automatically adapted to the evolution of the electric potential profiles. If, in the course of the simulation, strong changes of the dimensionless electric potential with x are detected in any x coordinate region, more grid points are added into that region. Appropriate simulation space grids were calculated in this way to ensure good accuracy.
- I also have doubts about problem formulation. Authors mix ionic concentrations and chemical potential. It is very hard to deduce what they really solve for. Moreover, normalization with the time-changing values (eq. 3 and eq. 5) is confusing. It would be very useful to write down full PNP equations for ionic concentrations and boundary conditions to understand better whether new terms appear there due to differentiations. The full PNP problem means all c+, c- and \Psi equations and boundary conditions without using \mu. In such a way, the picture will be more transparent.
Good point! The reason for the use of the ionic concentrations together with the electrochemical potential is that it readily allows to be extended to the case of finite ionic sizes. It is certainly confusing in the considered case of point ions so that we made the requested modifications in the revised version of the manuscript.
- Another major concern is that nowhere is the validity of the PNP approximation discussed.
A paragraph noting the limitations of the standard electrokinetic model has been added at the beginning of the Equation System section.
- Figures suggest that potentials of 20fold thermal energy are applied, which requires at least some discussion.
We agree. The following comment was added at the beginning of the discussion of Fig. 1:
Note that this being a theoretical work, the dimensionless surface potentials in this and the following Figures extend to very high values. While such values of the order of 400 mV are usually unrealistic, they were used in order to better understand the qualitative behavior of the plotted magnitudes.
- Here it is worth pointing also the notation. Authors seem to choose small y for dimensionless potential instead of Psi with an overbar, which they did for other variables. This is also confusing.
The notation was modified so that all the dimensionless variables are now marked by an overbar including the dimensionless potential Psi.
- Authors need to thoroughly address those issues before publication.
We thank Reviewer 1 for his thoughtful comments that helped improve the manuscript.
Reviewer 2 Report
Report on the manuscript 246386 Impedance-Frequency Response of Closed Electrolytic Cells by J.J. Lopez-Garcia et al, submitted for publication on micromachines The goal of the authors is to determine the ionic distribution and the electric response of an electrolytic cell submitted to an external bias plus a harmonic perturbation of small amplitude. The same problem was considered a few years ago in Ref.[40], where the main results obtained in the present manuscript were partially already discussed. The authors obtain, in the simple case where the positive and negative ions have the same diffusion coefficient, approximated formulae that can be useful in the interpretation of experimental data. The paper is well written, and could be of interest for searchers working in the impedance spectroscopy area. Unfortunately, in the present version, the references are not correctly written, and it is difficult to follow what has been done, and from who. It is mandatory to check the references and amend the quotation in the text. The first references are absent, or not correctly numbered. I marked on the pdf submitted by the authors to the journal a few points that need to be discussed before acceptance. In particular it is important top stress what of the quantities investigated in details in the manuscript, as the thickness of the double layer, are experimentally accessible. Also a discussion on the open system could be in order. In fact, in such a system, the bulk density of the ions in the center of the cell is fixed by diffusion from the reservoir, that is related to the diffusion of ions towards to the electrodes.

Author Response
The referees' comments and suggestions (regular font) and our answers (italics) follow:
Reviewer-2
Comments and Suggestions for Authors
- Report on the manuscript 246386 Impedance-Frequency Response of Closed Electrolytic Cells by J.J. Lopez-Garcia et al, submitted for publication on micromachines. The goal of the authors is to determine the ionic distribution and the electric response of an electrolytic cell submitted to an external bias plus a harmonic perturbation of small amplitude. The same problem was considered a few years ago in Ref.[40], where the main results obtained in the present manuscript were partially already discussed. The authors obtain, in the simple case where the positive and negative ions have the same diffusion coefficient, approximated formulae that can be useful in the interpretation of experimental data. The paper is well written, and could be of interest for searchers working in the impedance spectroscopy area. Unfortunately, in the present version, the references are not correctly written, and it is difficult to follow what has been done, and from who. It is mandatory to check the references and amend the quotation in the text. The first references are absent, or not correctly numbered.
The problem with the references is a result of the automatic transformation of our submitted manuscript to the Micromachines format. This has been solved in the revised manuscript.
- I marked on the pdf submitted by the authors to the journal a few points that need to be discussed before acceptance. In particular, it is important top stress what of the quantities investigated in details in the manuscript, as the thickness of the double layer, are experimentally accessible.
Points marked on the submitted pdf:
- Page 1. Already discussed in J. Mol. Liq. 272, 565, 2018
A comment has been included in page 15.
- Page 1. This assumption has to be proved because in the dynamical state the diffusion of the ions coming from the reservoir, and fixing the bulk density, is also frequency dependent.
While we agree with this remark, we think that this correction should be negligible since it depends on concentration changes due to the small AC potential amplitude compared to those determined by the large DC amplitude. We slightly modified the Abstract and added a comment after Eq. (7)
- Page 1. why?
Because, as noted in references [28] and [29], in many micro and nano devices there is no "infinite reservoir" required to satisfy the open cell condition.
- Page 7. According to me some information on the program to solve the PNP-equations can help the reader.
The following paragraph was added to the manuscript in order to clarify this point:
The numerical calculations were performed using an algorithm based on the network simulation method, which consists in modeling a physical process by means of a graphical representation analogous to circuit electrical diagrams, which is analyzed using an electric circuit simulation program such as PSpice. A full account of the network model used in this work is given in Ref. [5]. It must be noted that since the electric potential changes rapidly near the interface x=+-L, an appropriate simulation space grid must be modeled. In this work, the x-space grid is automatically adapted to the evolution of the electric potential profiles. If, in the course of the simulation, strong changes of the dimensionless electric potential with x are detected in any x coordinate region, more grid points are added into that region. Appropriate simulation space grids were calculated in this way to ensure good accuracy.
- Page 7. Moreover, is it possible to compare the theoretical predictions with some experimental data?
This is a theoretical work intended to show that the interpretation of experimental data using the usual open cell model could be misleading when the measurement cell is actually of the closed type. We did not attempt at this stage to make comparisons with experimental data.
- Page 10. Is this quantity of experimental importance?
We believe that it is important since the dependence of the double layer thickness with the DC electrode potential determines the shift of the high frequency dispersion as shown in the Frequency Response section.
- Page 11. What does it mean "high potential"?
In this context, "high potential" refers to dimensionless surface potential values much larger than unity that allow to simplify hyperbolic functions as shown in Appendix 1.
More precisely, equations (23) and (24) show that closed cell effects become important when a>>1. Therefore, for L/lambda_D=100 a potential value y_s>>9 would be "high" while for L/lambda_D=100 a value y_s>5 would be sufficient.
- Page 16. In the linear limit
A comment has been included in page 16.
Also a discussion on the open system could be in order. In fact, in such a system, the bulk density of the ions in the center of the cell is fixed by diffusion from the reservoir, that is related to the diffusion of ions towards to the electrodes.
A brief discussion of the open system, including an expression for the increment of the average ionic concentration in the cell required to maintain unaltered the concentration at its central plane when a potential difference is applied, Eq. (29), was already presented in the manuscript. This was slightly expanded in the revised version.
We thank Reviewer 2 for his pointed comments that contributed to improve the manuscript.

Round 2
Reviewer 1 Report
The authors made considerable efforts to make their manuscript clear and readable. I appreciate it very much.
I do have further comments relating to both merit and presentation, which I think are worth considering before the manuscript is published:
Part A. Presentation
1) The figures are nice and colourful yet should be prepared with a little more attention towards colours and content.
Anytime I tried to read the pictures I had a hard time distinguishing between short and long size limits as both were in black.
The colours can be correlated with the value presented in the legend. This will save a lot of effort for the reader.
Maybe it is worth considering making fewer lines (line 398, page 15) and also changing their textures, e.g. by introducing various line-point markers to make the presentations easier to read.
2) This is my personal opinion: I prefer when all legends are present in the figure or in the caption instead of looking for them in the text as one has to do regarding dots, dots-lines etc.
3) Figure 4 is discussed in the text in an awkward order: 4b 4a 4c. Judging by the disscussion of fig 5, it is
possible to improve it.
4) In eq. 46 authors add the displacement current. I am not an expert, so judging by the lack of comments around this, it is a standard procedure in this community. However, as not an expert, I would appreciate a comment sentence or two and a reference to why it is done. Let me explain the reason for my concern. The PNP equation is an approximation describing the motion of ions in a so-called quasi-static regime. There is an assumption about the time scales: The electrodynamic fields are quickly relaxing compared to the electrostatic fields that remain and drive the ionic motion. Adding the displacement current clearly states this is not the case. In such circumstances, PNP might be invalid, and its electromagnetic counterpart - MHD equations, might be necessary to consider. An appropriate comment would ease the doubts of a reader not used to the approximations made in this particular community.
Part B. Merit
1) Typically, the perturbation series are done with respect to some controllable parameter. Please refer to these two publications, where a similar "formulation" of the problem step is being done. This might help other researchers to navigate over approaches to AC series expansion:
--> A. Bandopadhyay, V. A. Shaik, and S. Chakraborty, Effects of nite ionic size and solvent polarization on the dynamics of electrolytes probed through harmonic disturbances, Phys. Rev. E 91 , 042307 (2015)
--> R. F. Stout and A. S. Khair, Moderately nonlinear discharge dynamics under an ac voltage, Phys. Rev. E 92, 032305 (2015).
These can also help authors to identify small parameters they perturb into and when this perturbation is valid.
2) Eq. 46 strongly assumes that the current is constant over the whole segment, which is true only in the stationary state or maybe in some other very special cases. In general, at any time instance, current can vary from place to place in the system, especially in the AC forcing. There we can expect some stationary-like arguments in the time average sense.
3) The authors claim multiple times (figure 4b and 5b. line 342 page 13, line 410 page 15) that there is a strong connection between characteristic frequency (eq. 60) and their results. I will make the following remark and I will support it with the evidence presented by the authors:
--> The time scales in the oscillatory problem involve a) AC forcing frequency b) diffusion over segment c) diffusion over Debye length. But the Debye length for the linear AC problem does not correspond to the initial concentration at the centre of the system but rather to c(0), which the authors calculate. That can be observed by the frequency dependence authors manifest on plots 4 and 5. Once recalled the maximum should be at the same point, where forcing frequency is comparable with the diffusion time scale on the Debye length calculated based on c(0)
--> Such a scaling done alone in the \omega domain will change the appearance of the figures, but it can be restituted by similarly scaling the X axis. According to the formulas 50-52, the Debye length also appears there.
I expect that and paper will have a great benefit if the authors can show it to the reader in a convincing way.
3) It appears to me that both Figures 4a (black and orange curves) and 5c contain nonmonotonicities with parameter present on the legend, which authors should comment on.
Once these claims are adressed, I will be very happy to support the publication.
Author Response
The referees' comments and suggestions (regular font) and our answers (italics) follow:
The authors made considerable efforts to make their manuscript clear and readable. I appreciate it very much.
I do have further comments relating to both merit and presentation, which I think are worth considering before the manuscript is published:
Part A. Presentation
1) The figures are nice and colourful yet should be prepared with a little more attention towards colours and content.
Anytime I tried to read the pictures I had a hard time distinguishing between short and long size limits as both were in black.
Good point! In all the figures we replaced the dark blue color by cyan.
The colours can be correlated with the value presented in the legend. This will save a lot of effort for the reader.
The requested change has been made.
Maybe it is worth considering making fewer lines (line 398, page 15) and also changing their textures, e.g. by introducing various line-point markers to make the presentations easier to read.
The number of lines in Fig. 4c was reduced to match Fig 5c.
2) This is my personal opinion: I prefer when all legends are present in the figure or in the caption instead of looking for them in the text as one has to do regarding dots, dots-lines etc.
The missing information was added to the captions of Figs. 3, 4, and 5.
3) Figure 4 is discussed in the text in an awkward order: 4b 4a 4c. Judging by the disscussion of fig 5, it is possible to improve it.
This awkwardness has been solved.
4) In eq. 46 authors add the displacement current. I am not an expert, so judging by the lack of comments around this, it is a standard procedure in this community. However, as not an expert, I would appreciate a comment sentence or two and a reference to why it is done. Let me explain the reason for my concern. The PNP equation is an approximation describing the motion of ions in a so-called quasi-static regime. There is an assumption about the time scales: The electrodynamic fields are quickly relaxing compared to the electrostatic fields that remain and drive the ionic motion. Adding the displacement current clearly states this is not the case. In such circumstances, PNP might be invalid, and its electromagnetic counterpart - MHD equations, might be necessary to consider. An appropriate comment would ease the doubts of a reader not used to the approximations made in this particular community.
This must be done even at low frequencies when the faradaic current varies along the length over which the impedance is being calculated, in view of the continuity equation.
A note and a reference to:
[33] Cohen, H.; Cooley, J.W. The numerical solution of the time-dependent Nernst-Planck equations, Biophysical J. 1965, 5, 145-162.
have been added to the manuscript after Eq. (46).
Part B. Merit
1) Typically, the perturbation series are done with respect to some controllable parameter. Please refer to these two publications, where a similar "formulation" of the problem step is being done. This might help other researchers to navigate over approaches to AC series expansion:
--> A. Bandopadhyay, V. A. Shaik, and S. Chakraborty, Effects of nite ionic size and solvent polarization on the dynamics of electrolytes probed through harmonic disturbances, Phys. Rev. E 91 , 042307 (2015)
--> R. F. Stout and A. S. Khair, Moderately nonlinear discharge dynamics under an ac voltage, Phys. Rev. E 92, 032305 (2015).
These can also help authors to identify small parameters they perturb into and when this perturbation is valid.
Good point! We added the following comment including the suggested references after Eq. (31):
Note that this is a strictly linear expansion leading to an AC current proportional and a cell impedance independent of the AC voltage, unlike higher order expansions where these assumptions do not hold [33,34].
2) Eq. 46 strongly assumes that the current is constant over the whole segment, which is true only in the stationary state or maybe in some other very special cases. In general, at any time instance, current can vary from place to place in the system, especially in the AC forcing. There we can expect some stationary-like arguments in the time average sense.
The assumption of a constant total current value over the whole segment is precisely the reason for adding the displacement current term. An example is a RC circuit in which the total (faradaic plus displacement) current across the resistor has the same value as across the condenser.
By the way, . Bandopadhyay, V. A. Shaik, and S. Chakraborty, use the displacement current term in their Eq. (14).
3) The authors claim multiple times (figure 4b and 5b. line 342 page 13, line 410 page 15) that there is a strong connection between characteristic frequency (eq. 60) and their results. I will make the following remark and I will support it with the evidence presented by the authors:
--> The time scales in the oscillatory problem involve a) AC forcing frequency b) diffusion over segment c) diffusion over Debye length. But the Debye length for the linear AC problem does not correspond to the initial concentration at the centre of the system but rather to c(0), which the authors calculate. That can be observed by the frequency dependence authors manifest on plots 4 and 5. Once recalled the maximum should be at the same point, where forcing frequency is comparable with the diffusion time scale on the Debye length calculated based on c(0)
Sorry, we do not understand this objection. The characteristic frequency of the high frequency dispersion does indeed depend on the concentration c(0) since the AC signal is applied to the stationary DC state rather than the equilibrium state with no bias. The vertical dotted lines in Fig. 5b, for example, show a remarkable agreement with Eq.(60).
--> Such a scaling done alone in the \omega domain will change the appearance of the figures, but it can be restituted by similarly scaling the X axis. According to the formulas 50-52, the Debye length also appears there.
I expect that and paper will have a great benefit if the authors can show it to the reader in a convincing way.
3) It appears to me that both Figures 4a (black and orange curves) and 5c contain nonmonotonicities with parameter present on the legend, which authors should comment on.
The non monotonous behavior of the black (actually dark blue) curve in Fig. 4a is commented in the paragraph following Eq. (60). The behavior of the orange curve is still monotonous.
The non monotonous behavior in Fig. 5c is discussed in the last paragraph before the Conclusions.
Once these claims are adressed, I will be very happy to support the publication.
We thank the Referee for his pointed and helpful comments that contributed to improve the manuscript.
